# Effects of Pre-Collegiate Sport Specialization on Cognitive, Postural, and Psychological Functions: Findings from the NCAA-DoD CARE Consortium

**DOI:** 10.3390/ijerph19042335

**Published:** 2022-02-18

**Authors:** Tsung-Yeh Chou, Jaclyn B. Caccese, Yu-Lun Huang, Joseph J. Glutting, Thomas A. Buckley, Steven P. Broglio, Thomas W. McAllister, Michael A. McCrea, Paul F. Pasquina, Thomas W. Kaminski

**Affiliations:** 1Department of Kinesiology and Applied Physiology, University of Delaware, 547 South College Avenue, Newark, DE 19716, USA; tsungyeh@udel.edu (T.-Y.C.); tbuckley@udel.edu (T.A.B.); kaminski@udel.edu (T.W.K.); 2School of Health and Rehabilitation Science, College of Medicine, The Ohio State University, 453 W 10th Ave, Columbus, OH 43210, USA; jaclyn.caccese@osumc.edu; 3Department of Physical Education and Sport Science, National Taiwan Normal University, No. 162, Sec. 1, Heping E. Rd., Da’an Dist., Taipei City 106, Taiwan; 4School of Education, University of Delaware, 106 Alison Hall West, Newark, DE 19716, USA; glutting@udel.edu; 5Michigan Concussion Center, School of Kinesiology, University of Michigan, 830 N University Ave, Ann Arbor, MI 48109, USA; broglio@umich.edu; 6Department of Psychiatry, School of Medicine, Indiana University, 340 West 10th Street Fairbanks Hall, Suite 6200, Indianapolis, IN 46202, USA; twmcalli@iupui.edu; 7Department of Neurosurgery, Medical College of Wisconsin, 8701 Watertown Plank Road, Milwaukee, WI 53226, USA; mmccrea@mcw.edu; 8Department of Physical Medicine and Rehabilitation, Uniformed Services University, 4301 Jones Bridge Road, Bethesda, MD 20814, USA; paul.pasquina@usuhs.edu

**Keywords:** sport sampling, youth sport, balance, mental health, cognition

## Abstract

Background: Early sport specialization has been associated with an increased risk of musculoskeletal injuries and unfavorable psychological outcomes; however, it is unknown whether sport specialization is associated with worse cognitive, postural, and psychological functions in first-year collegiate student-athletes. Methods: First-year collegiate multisport (MA) and single-sport (SA) student-athletes were identified using a pre-collegiate sport experience questionnaire. The cognitive, postural, and psychological functions were assessed by the Immediate Post-Concussion Assessment and Cognitive Testing (ImPACT), Standardized Assessment of Concussion (SAC), Balance Error Scoring System (BESS), and Brief Symptom Inventory 18 (BSI-18). Results: MA student-athletes performed higher in cognitive outcomes (e.g., higher ImPACT visual memory composite scores [ß = 0.056, *p* < 0.001]), but had higher psychological distress (e.g., higher BSI-18 global severity index [ß = 0.057, *p* < 0.001]) and no difference in postural stability (*p* > 0.05) than SA student-athletes. Conclusions: This study indicated first-year collegiate athletes with a history of sport specialization demonstrate lower cognitive performance but decreased psychological distress and no differences in static postural stability as compared to their MA counterparts. Future studies should consider involving different health measures to better understand the influence of sport specialization on overall physical and mental health.

## 1. Introduction

In the United States, the youth sport population has grown close to 30 million [1]. Youth sports introduce children to the importance of a healthy lifestyle, social networking, and sport enjoyment [2]. Recently, sport specialization has been introduced to facilitate skill acquisition and proficiency in a given sport and assist participants in achieving elite-level performance [3]. However, there are rising concerns about sport specialization, defined as the year-round intensive training of a single sport to the exclusion of other sports [3,4]. Early sport specialization may potentially affect many aspects of youth health including both physical and emotional characteristics. Early evidence suggests that youth single-sport student-athletes (SAs) are at higher risk for burnout and psychological stress when compared with their multisport student-athlete (MA) counterparts [5,6]. Additionally, SAs are more likely to report a previous injury in the past 12 months [7] (odds ratio = 1.59), especially an overuse injury [4] (odds ratio = 2.25) as compared to athletes with lower degrees of sport specialization. Similarly, student-athletes with high training volume (i.e., >17.5 h/week), usually observed in SAs, have poor self-reported psychological well-being [8] and altered sleep patterns, both of which have been associated with poor academic performance [9]. Furthermore, SAs have demonstrated poorer postural stability [10] and more aberrant landing strategies [11,12] than MAs. It seems possible that SAs are likely to have insufficient recovery following repetitive use of the same muscle groups, which is further complicated by the development of improper neuromuscular techniques, greater athletic exposure, and high-intensity training [13]. Furthermore, the time devoted to a single sport may limit their social interaction and thus alter relationships with their peers [14]. Therefore, playing multiple sports, also known as sports sampling, is recommended for youth athletes [6,15,16].

Although the mechanisms of an increase in injury risk and psychological distress among sport-specialized athletes remains unknown, participating in many sports while growing up provides youth with opportunities to develop fundamental motor skills, social interaction, and overall psychological and musculoskeletal health [6,17]. In fact, multisport experience is associated with enhanced neuromuscular control in jump-landing biomechanics [11,12], better self-regulated emotional skills [16], and smoother social interactions with their peers and adults [5]. To date, the majority of studies involving sport specialization have investigated musculoskeletal injury risk [7,18,19] and specific psychological outcomes among high-school student-athletes [5,6]. However, it is unknown whether a history of multisport participation also promotes enhancements in cognitive, postural, and psychological functions in collegiate student-athletes that may provide novel and meaningful clinical information related to their overall health and well-being.

To better understand the association between a variety of measures of physical and emotional health and youth sport specialization, we accessed data from a large database of collegiate athletes to help answer our questions. Therefore, the purpose of this study was to investigate the effect of pre-collegiate sport specialization on neuropsychological cognitive function, balance, and emotional health measurements acquired from a large cohort of intercollegiate athletes. We hypothesized that a history of sport specialization would be associated with worse cognition, increased psychological distress, and poor balance performance in first-year collegiate student-athletes.

## 2. Materials and Methods

The present study was part of the large concussion research investigation involving 30 universities across the United States, titled the National Collegiate Athletic Association (NCAA)-Department of Defense (DoD) Grand Alliance: Concussion Assessment, Research, and Education (CARE) Consortium [20]. To determine the effect of pre-collegiate sport specialization on cognitive, postural, and psychological function, we performed secondary data analysis using data from the baseline concussion assessment battery [20], including the Immediate Post-Concussion Assessment and Cognitive Test (ImPACT Applications, Inc.), Standardized Assessment of Concussion (SAC), Balance Error Scoring System (BESS), Brief Symptom Inventory 18 (BSI-18), and a self-reported questionnaire consisting of demographic information, medical history, and sport history forms from athletes enrolled between January 2014 and December 2017 [20]. The study procedures were approved by the University of Michigan Institutional Review Board, the US Army Medical Research and Development Command’s (USAMRDC) Office of Research Protections, the Human Research Protection Office (HRPO), and each local institutional review board.

### 2.1. Participants

Participants were first-year (i.e., freshmen) NCAA student-athletes from 26 sports [20]. All participants completed a questionnaire including demographic information (e.g., age, sex, current academic year, race/ethnicity), medical history (e.g., concussion history, learning disorders, attention deficit hyperactivity disorder (ADHD)), and a sport history form to collect pre-collegiate sport participation history. All student-athletes, and their parent/guardian if younger than 18 years, signed written informed consent prior to assessment. This study was completed in accordance with the Declaration of Helsinki. The inclusion criteria of this study are student-athletes who (1) completed the assessments prior to the beginning of their competitive sport season, (2) reported their current academic year was freshman, and (3) matched our MA or SA criteria. Student-athletes who were non-NCAA athlete athletes (e.g., non-sport military cadets) were excluded.

We utilized a self-reported pre-collegiate sport participation form to categorize student-athletes into the MA and SA groups. The form had three questions to assess sports participation history: (1) “From birth, how many years have you participated in your primary sport?” (2) “From birth, have you participated in any other organized sports?” (3) “From birth, indicate if you have participated in any of the following organized sports (other than your primary sport), and also indicate the number of years of participation for each sport”. We utilized a minimum of three years of sport participation as the criterion for our MA and SA group assignment. Student-athletes who answered “No” to question two and had at least three years of participation in their primary sport before their collegiate career were categorized as SAs. Student-athletes who answered “Yes” to question two and had at least three years of participation with two or more sports before their collegiate career were categorized as MAs. If their responses to additional sport participation (question 3) were “one year” or “two years”, they were excluded from our data analysis because several years of sport participation are essential to acquire adequate motor skills in a particular sport [21]. For example, a soccer player with a minimum of three years in soccer, and that has no other sport participation history was classified into the SA group. Conversely, a soccer player with a minimum of three years in soccer and with a minimum of three years in another sport(s) (e.g., three years of basketball or three years of basketball and two years of baseball) was classified into the MA group. Student-athletes who listed similar sports such as volleyball and beach volleyball, cross country and track, or swimming and diving would be categorized as participating in one sport.

### 2.2. Outcome Measures

All outcome measures were obtained by the respective research teams at the CARE sites and in a controlled environment. The ImPACT is a widely used computerized assessment to evaluate cognitive function [22]. It generates four composite scores, including verbal memory (VEM), visual memory (VIM), visual motor speed (VMS), and reaction time (RT). Higher composite scores indicate better cognitive performance, except for the RT score. The ImPACT has good validity [23], but low to moderate test–retest reliability [24,25]. The BSI-18, a self-reported assessment of psychological distress, contains 18 questions rated on a scale from 0 to 5. Participants were asked to rate their level of distress in the past 7 days. Outcomes include the Global Severity Index (GSI) and three sub-scores including somatization, depression, and anxiety. A higher score indicates greater psychological distress [26]. The BSI-18 has demonstrated good internal consistency and validity in measuring psychological functioning in high school and collegiate athletes [27]. The SAC is a cognitive screening tool that includes measures of orientation, immediate memory, concentration, and delayed recall for a total score range from 0 to 30 [20]. Higher scores indicate better cognitive function. The SAC has demonstrated good validity and reliability on concussion assessment among athletic populations [28]. Lastly, the BESS is a clinical measure of static postural stability, involving measurements from three different stances (i.e., single-limb, double-limb, and tandem) on two surfaces (i.e., firm, foam) for a total of 6 conditions [20]. Higher scores indicate worse postural stability [20]. The interrater and intrarater reliability for the BESS total scores were 0.57 and 0.74, respectively [29].

### 2.3. Statistical Analysis

Direct-entry multiple regression analyses (MRAs) were conducted to investigate the association between a history of sport specialization and all outcome measures. The independent variable was sport specialization (MA and SA), while the dependent variables included the four ImPACT composite scores, SAC and BESS total scores, and BSI-18 GSI score along with the three sub-scores for somatization, depression, and anxiety. Based on a previous study [30], covariates, including sex, race/ethnicity, learning disorders, ADHD, and previous concussion history, were adjusted in the model.

The multiple regression analyses were conducted independently for each of the 10 dependent variables. The predictor in the models was sport specialization (MA vs. SA) and covariates including sex (male vs. female), previous concussion history (yes vs. no), learning disorders (yes vs. no), ADHD (yes vs. no), and race/ethnicity (Caucasian/non-Hispanic, African American, other race/ethnicity). Because race/ethnicity included three groups, two dummy variables were created (Caucasian/non-Hispanic vs. African American, Caucasian/non-Hispanic vs. other race/ethnicity). The $R^{2}$ value served as the effect size of the linear combination of the predictor and covariates, where $R^{2}$ values of 0.02 represent a small effect, 0.13 represent a medium effect, and 0.25 represent a large effect [31]. The standardized beta coefficient (ß) was utilized to explain the contribution of one predictor to the amount of change in outcome measures, where ß values of 0.10 represent a small effect, 0.20 represent a medium effect, and 0.30 represent a large effect. Bootstrapping statistical techniques were conducted for the models that were not normally distributed [32]. All of our multiple regression analyses were completed using SPSS Version-26 (IBM, Armonk, NY, USA). The level of significance was set a priori at *p* < 0.05. According to the recommendations of previous literature, the sample size per independent variable was adequate for each regression analysis [33]. To evaluate the multicollinearity, we set the threshold of the Variance Inflation Factor (VIF) at 10 [34]. After reviewing the VIF values (range = 1.02–1.06) for each regression analysis, multicollinearity did not present in the regression model.

## 3. Results

The descriptive information for the demographic data and sport category are listed in Table 1 and Table 2, respectively.

There were 8987 student-athletes who met the inclusion criteria; however, 516 were excluded due to missing data. Thus, 8471 first-year student-athletes (MA = 5410, SA = 3061) were included in the analyses (Figure 1).

The MRA models for all outcome measures were statistically significant (*p* < 0.001) for our predictor (Table 3). However, only six outcome measures reached a small effect size: SAC total score ($R^{2}$ = 0.025), BSI-18 GSI ($R^{2}$ = 0.021), VEM ($R^{2}$ = 0.023), VIM ($R^{2}$ = 0.021), VMS ($R^{2}$ = 0.046), and RT composite scores ($R^{2}$ = 0.045) (Table 3).

Our primary-interest predictor (sport specialization) contributed significantly to 7 out of 10 outcome measures (Table 4), suggesting that MAs had higher SAC total score (ß = 0.042), ImPACT VIM (ß = 0.056), VMS (ß = 0.075), and lower RT composite score (ß = −0.094) values than SAs. Conversely, MAs had higher BSI-18 GSI (ß = 0.057), BSI-18 somatization (ß = 0.067), and BSI-18 anxiety sub-score (ß = 0.049) values than SAs. There were no differences in BESS total scores.

## 4. Discussion

Historically, researchers have examined the relationship between youth sport specialization and musculoskeletal injury risk [4,7,18] and neuromuscular control [11,12]. The present study was the first study to investigate the effects of pre-collegiate sport specialization on measures of cognitive, postural, and mental health in a large group of intercollegiate student-athletes. We hypothesized that MAs would perform better than SAs on all outcomes. The results suggest that the MA group performed better in cognitive function but exhibited greater psychological distress and no difference in postural stability. In cases when statistical significance was observed, the small effect sizes suggested the differences were minimal (e.g., <2-point difference in VIM composite score).

The MA group reported greater psychological distress, which is inconsistent with previous literature suggesting psychological benefits arise from multisport experiences in collegiate athletes [35]. Garinger et al. [35] reported collegiate athletes participating in multiple sports demonstrated lower levels of psychological stress compared to athletes who participated in a single sport. Their findings were different from the present study. A possible explanation of this conflicting result is that Garinger et al. [35] assessed perfectionism and burnout, but we assessed psychological distress related to depression, anxiety, and somatization. Lancaster and colleagues [27] reported that athletes with multiple concussion histories tended to report higher (worse) BSI-18 GSI scores when compared to those without a history of concussion. While controlling for the concussion history in our analysis, the MA group reported higher BSI-18 scores, suggesting the effect of sport participation history was independent of their concussion history. It is possible that the MA group presented with higher psychological distress due to the transition period of specializing in a single sport at the collegiate level. We may have seen a larger effect of sport specialization if the survey was administered during the mid-season, as the BSI-18 asks participants to report symptoms experienced in the past week. Overall, the scores in both groups fell within the lower percentile of the reported value, suggesting less psychological stress among all in both groups [27].

A growing body of evidence suggests a positive connection between exercise and cognitive function in youth aged 5–13 [36]. Children with higher fitness levels performed better in cognitive functions than children with lower fitness levels [37,38,39]. Cross-sectional studies that involved imaging techniques also observed a greater volume of brain structures that assist executive function [40] (e.g., caudate nucleus, putamen) and memory [41] (e.g., hippocampus) in children with higher aerobic fitness. Thus, it seems that exercise fosters the development of cognitive function during childhood and perhaps throughout adolescence if they continue to exercise. In our study, we proposed student-athletes participating in multiple sports prior to attending college would also exhibit the same benefit and potentially induce a greater effect. After adjusting the covariates, our findings suggest that the benefits of multisport participation on cognitive function are significant; however, the differences are small. It is important to note, when compared to published normative data involving cognitive and neurobehavioral assessments in other intercollegiate athletic populations, our MA and SA groups had comparable values in these measures [27,42,43], suggesting that sport specialization did not result in worse neuropsychological functioning overall or a lack of sensitivity of the selected measures in the present study.

Pre-collegiate sport specialization did not predict deficits or improvements in static postural measurements in the present study. The lack of statistical significance in the BESS total score may suggest that the static postural stability assessment is less affected by sport participation than dynamic postural measurements are. For example, greater dynamic postural instability, measured by the Y-balance test, has been observed in high school SAs compared to MAs [10]. Other measures of postural stability relative to functional movement patterns captured during athletic performance have been shown to be more sensitive to the effects of participating in single versus multiple sports [11,12]. Furthermore, advanced assessments of jump-landing biomechanics have demonstrated a greater increase in hip and knee extensor moments and a decrease in knee abduction angle and abduction moment in high school MAs compared with SAs [44]. Our participants were young and active NCAA student-athletes, who may have had a greater capacity to compensate or recover from static postural instability, thus making it difficult to distinguish substantial differences between groups. Therefore, assessments that focus on more athletic-related movement patterns may have provided a more challenging paradigm and generated different results. Future studies should consider involving functional tasks to evaluate postural stability in collegiate student-athletes with a history of sport specialization.

The concept of early sport specialization was developed from the conceptual theory proposed by Ericsson and colleagues based upon the 10,000 h or 10-year rule to achieve expert musical prowess [45]. Sport theorists later applied this model to youth athletes, and in order to accomplish sporting success, these athletes needed to start in the early years of development. Contemporary reports on early sport specialization in adolescent athletes [46,47,48] suggest that early exposure to a diversity of sport experiences and late specialization appear to be better predictors of achieving elite athletic status [49]. Several previous reports have demonstrated a number of psychosocial benefits in youth MAs. In contrast, many adverse effects have been reported in SAs including social isolation, maladaptive behaviors, risk of musculoskeletal injuries, and premature dropout/burnout [5,16,50]. Furthermore, early exposure (e.g., 6–12 years old) to multiple sports is presumed to offer a friendly psychosocial environment and more interpersonal experiences to facilitate personal development than a focus on a single sport endeavor [51]. Nonetheless, our findings suggest single-sport specialization does not result in altered cognitive function, greater psychological distress, or postural instability among first-year NCAA student-athletes. Clinicians may consider using a multifaced approach to evaluate student-athletes’ overall physical and mental health and make the appropriate resources available.

The results of the current study should be interpreted considering the following limitations. Our definitions of SA [3,13] and MA [11,12] were modified slightly from the previous literature. The self-reported sport history questionnaire was not designed to determine whether student-athletes were sport specialized or playing multiple sports before entering college; however, the SAs in the present study only participated in a single sport throughout their athletic career. Thus, the definition of SA in the current study would be more stringent than the commonly used definition and potentially provide more information on the effect of sport specialization. In addition, we chose the threshold of at least three years of sport participation because the self-reported sport history form did not allow the ability to discern overlapping sports participation. We would also like to suggest that a survival bias may exist within our participants because a small percentage of high school athletes advance to the collegiate level [52] (an average of 7% across all NCAA sports), and perhaps those possessing negative effects due to sport specialization quit before the collegiate level or developed career-ending injuries. Lastly, this study investigated the effect of sport specialization among NCAA student-athletes so our findings cannot be extended to other ages or levels of play.

## 5. Conclusions

Pre-collegiate sport specialization is associated with lower cognitive function but decreased psychological distress in first-year NCAA student-athletes; however, no difference in the measures of balance was observed. A prospective research design and the utilization of various health measures are warranted for future studies to understand the influence of sport specialization on overall physical and mental health.

## Figures and Tables

**Figure 1 ijerph-19-02335-f001:**
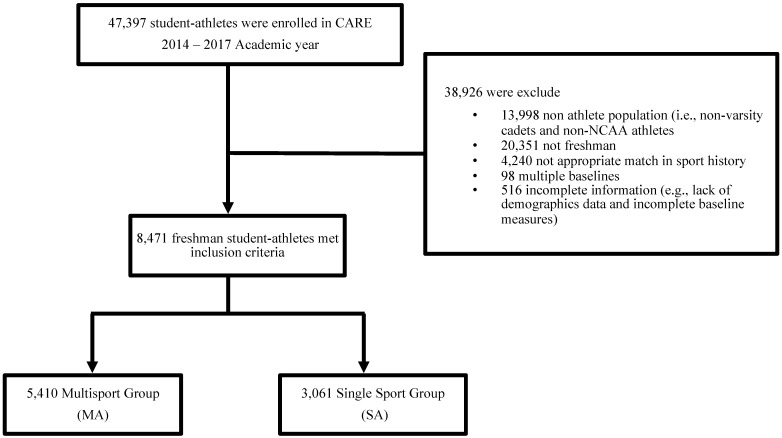
Description of participant selection.

**Table 1 ijerph-19-02335-t001:** Demographic information of MA and SA groups.

	MA Group N = 5410	SA Group N = 3061
Demographics		
Race, Caucasian, *n* (%)	4175 (77)	2047 (67)
Sex, female, *n* (%)	2091 (39)	1398 (46)
Age ± SD, year	18.2 ± 0.6	18.2 ± 0.7
Height ± SD, cm	178.4 ± 10.7	176.3 ± 11.5
Mass ± SD, kg	78.1 ± 17.8	74.8 ± 18.2
Self-reported history		
Additional sport participation ± SD, *n*	2.1 ± 1.2	N/A
Previous concussion, *n* (%)	1439 (27)	544 (18)
ADHD, *n* (%)	340 (6)	171 (6)
Learning disorders, *n* (%)	153 (3)	74 (2)

Data represent mean ± SD and number (%); Abbreviations: MA, multisport student-athlete; SA, single-sport student-athlete; SD, standard deviation; ADHD, attention deficit hyperactivity disorder; N/A, not applicable.

**Table 2 ijerph-19-02335-t002:** Frequency and primary sport categories identified among MA and SA.

	MA Group N = 5410	SA Group N = 3061
Non-contact sport		
Cheerleading, *n* (%)	62 (1.15)	80 (2.61)
Golf, *n* (%)	106 (1.96)	66 (2.16)
Rifle, *n* (%)	20 (0.37)	9 (0.29)
Rowing/Crew, *n* (%)	184 (3.4)	109 (3.56)
Swimming, *n* (%)	296 (5.47)	362 (11.83)
Sailing, *n* (%)	19 (0.35)	3 (0.10)
Tennis, *n* (%)	115 (2.13)	178 (5.81)
Limited contact sport		
Baseball, *n* (%)	436 (8.06)	176 (5.75)
Beach Volleyball, *n* (%)	6 (0.11)	4 (0.13)
Cross country/Track, *n* (%)	640 (11.83)	320 (10.45)
Fencing, *n* (%)	15 (0.28)	34 (1.11)
Field event, *n* (%)	185 (3.42)	46 (1.50)
Gymnastics, *n* (%)	44 (0.81)	158 (5.16)
Softball, *n* (%)	196 (3.62)	97 (3.17)
Volleyball, *n* (%)	186 (3.44)	113 (3.70)
Contact sport		
Basketball, *n* (%)	279 (5.16)	175 (5.71)
Diving, *n* (%)	60 (1.1)	28 (0.91)
Field hockey, *n* (%)	93 (1.72)	26 (0.85)
Football, *n* (%)	1196 (22.11)	410 (13.40)
Ice hockey, *n* (%)	93 (1.72)	38 (1.24)
Lacrosse, *n* (%)	394 (7.28)	82 (2.69)
Soccer, *n* (%)	507 (9.37)	388 (12.68)
Water polo, *n* (%)	96 (1.77)	55 (1.80)
Wrestling, *n* (%)	182 (3.36)	104 (3.40)

Abbreviations: MA, multisport student-athlete; SA, single-sport student-athlete.

**Table 3 ijerph-19-02335-t003:** The result of each outcome measure and the regression model among MA and SA groups.

	MA Group		SA Group		Regression Model		
	Mean	SD	Mean	SD	$R^{2}$	F	*p*
Outcome measure							
SAC	27.32	1.92	27.13	1.97	0.025 ^†^	28.391	<0.001
BESS	13.49	6.23	13.55	6.41	0.010	11.177	<0.001
BSI-18 Score							
GSI	3.11	5.37	2.54	5.29	0.021 ^†^	24.450	<0.001
Somatization	1.07	22.05	0.80	1.93	0.017	20.259	<0.001
Depression	0.94	2.11	0.83	2.05	0.009	10.451	<0.001
Anxiety	1.10	2.19	0.90	2.13	0.024 †	28.390	< 0.001
ImPACT composite Score							
Verbal memory	87.20	10.29	86.59	10.70	0.023 ^†^	20.789	<0.001
Visual memory	78.18	13.12	76.13	13.37	0.021 ^†^	19.031	<0.001
Visual motor speed	40.89	6.28	39.78	6.62	0.046 ^†^	43.049	<0.001
Reaction time (s)	0.60	0.09	0.62	0.11	0.044 ^†^	41.933	<0.001

Dependent variables in the regression model were the 10 outcome measures. Multiple correlation coefficient squared (*R*^2^): 0.02 = small effect size, 0.13 = medium effect size, and 0.25 = large effect size. ^†^ *R*^2^ with small effective size. Bootstrap results are based on 1000 bootstrap samples. Abbreviations: MA, multisport student-athlete; SA, single-sport student-athlete; SAC, standardized assessment of concussion total score; BESS, balance error scoring system total score; BSI-18, Brief Symptom Inventory 18; GSI, global severity index; ImPACT, Immediate Post-Concussion Assessment and Cognitive Testing. Higher score indicates better outcomes, except BESS, BSI-18, and ImPACT reaction time.

**Table 4 ijerph-19-02335-t004:** The effect size of predictor and covariates.

	Standardized Beta Coefficient (ß)						
	Sport Specialization	Sex	Concussion History	LD	ADHD	African American vs. Caucasian	Other Race vs. Caucasian
Outcome measure							
SAC	0.042 *	−0.52 *	-	−0.076 *	-	−0.110 *^,†^	-
BESS	-	0.087 *	-	-	-	-	−0.029
BSI-18 Score							
GSI	0.057 *	−0.110 *^,†^	0.056 *	0.027	0.024	-	-
Somatization	0.067 *	−0.093 *	0.064 *	-	-	-	-
Depression	0.029	−0.067 *	0.039	0.032	0.026	-	-
Anxiety	0.049 *	−0.119 *^,†^	0.041	0.024	0.036	−0.048 *	-
ImPACT composite Score							
Verbal memory	-	−0.053 *	0.033	−0.050 *	−0.066 *	−0.092 *	-
Visual memory	0.056 *	0.077 *	-	−0.040	−0.033	−0.087 *	-
Visual motor speed	0.075 *	−0.031	0.038	−0.113 *^,†^	−0.027	−0.134 *^,†^	0.027
Reaction time	−0.094 *	-	−0.38	0.045 *	0.039	0.163 *^,†^	-

The predictor was sport specialization (MA vs. SA). Covariates included sex (male vs. female), previous concussion history (yes vs. no), learning disorders (yes vs. no), ADHD (yes vs. no), and race/ethnicity (Caucasian/non-Hispanic, African American, other race/ethnicity). Standardized beta coefficients (ß): 0.10 = small effect size, 0.30 = medium effect size, and 0.50 = large effect size. * *p* < 0.001. ^†^ Beta coefficients with small effect size. Bootstrap results are based on 1000 bootstrap samples. Abbreviations: LD, learning disorders; ADHD, attention deficit hyperactivity disorder; SAC, standardized assessment of concussion total score; BESS, balance error scoring system total score; BSI-18, Brief Symptom Inventory 18; GSI, global severity index; ImPACT, Immediate Post-Concussion Assessment and Cognitive Testing.

## Data Availability

The CARE Consortium datasets generated and analyzed during the current study are available in the FITBIR repository (https://fitbir.nih.gov/ accessed on 14 February 2022).

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
