# Peer review of "Effects of Pre-Collegiate Sport Specialization on Cognitive, Postural, and Psychological Functions: Findings from the NCAA-DoD CARE Consortium"

_ijerph, 2022, doi:10.3390/ijerph19042335_

Round 1

Reviewer 1 Report

 Effects of pre-collegiate sport specialization on cognitive, postural, and psychological functions: Findings from the NCAADoD CARE Consortium

This paper explores aspects of early sport specialization which have been associated with an increased risk of musculoskeletal injuries and unfavorable psychological outcomes. Methodologically, certain individual differences were measured [the Immediate Post-Concussion Assessment and Cognitive Test (ImPACT Applications, Inc.), Standardized Assessment of Concussion (SAC), Balance Error Scoring System (BESS), Brief Symptom Inventory (BSI-18), and self-reported questionnaire consisted with demographic information, medical history, and sport history forms], and analyzed via multiple regression analyses. The results showed that the first-year collegiate athletes with a history of sport specialization demonstrate lower cognitive performance, but decreased psychological distress, while no differences in static postural stability compared to other counterparts. Discussion on the finding and future studies are provided.

The study presented in this paper is interesting and the results are useful, while the paper is well written. In my opinion, the paper is worth publishing after some minor revision which concerns, mainly some methodological issues.

-  -Please provide a short section with the limitation of the study.

 On the multiple regression analyses used:

-R2 values should be depicted on the Tables.

-Are the number of variables in the multiple regression analyses quite large compare to the sample size?

-Are there any multicollinearity effects present that affect the liner model?

-In the table caption, it is not clear which is the dependent variable

Author Response

1. Please provide a short section with the limitation of the study.

Thank you for your valuable feedback. We provided a paragraph of the limitation of the study in the last section of the discussion. Please refer to the section on line 324 - 339.

2. R2 values should be depicted on the Tables.

Thank you for your constructive comments. Since the reviewer did not specify which table we should include the R^2 values, we were unable to revise tables according to the feedback. However, we have provided the R^2 value for each regression model in table 3. Please refer to table 3 for more details. We also provided the R^2 value on line 200 - 201.

3. Are the number of variables in the multiple regression analyses quite large compare to the sample size?

Thank you for your constructive comments. In the current study, we included seven independent variables and 8,471 participants. Based on the previous study [1], the minimum required sample size for linear regression analysis was 200, and the recommended number of subjects per variable was 20. Therefore, the statistical analyses in the present study should have an adequate sample size to the number of variables. We also add more information in the statistical analysis section regarding the concerns of sample size. Please refer to line 178 - 179.

  1. Austin PC, Steyerberg EW. The number of subjects per variable required in linear regression analyses. J Clin Epidemiol [Internet]. Elsevier Inc; 2015;68:627–36. Available from: http://dx.doi.org/10.1016/j.jclinepi.2014.12.014

4. Are there any multicollinearity effects present that affect the liner model?

Thank you for your valuable feedback. To evaluate the multicollinearity in the present study, we set the threshold of Variance Inflation Factor (VIF) at 10 [2]. After reviewing the VIF values (range = 1.02 – 1.06) for each regression analysis, the multicollinearity did not present in the regression model. We also add more information in the statistical analysis section regarding the concerns of multicollinearity. Please refer to line 179 - 182.

  1. Vittinghoff E, Glidden DV, Shiboski SC MC. Regression Methods in Biostatistics: Linear, Logistic, Survival, and Repeated Measures Models. 2nd ed. Springer; 2012.

5. In the table caption, it is not clear which is the dependent variable

Thank you for your constructive comments. Since the reviewer did not specify which table caption did not provide clear information regarding the dependent variable, we could not revise tables according to the feedback. However, we have added one sentence to clarify the dependent variable for each regression model in table 3. Please refer to table 3 for more details. We also provided the dependent variable information on line 159 - 161.

Reviewer 2 Report

The authors evaluated the effects of sport specialization on psychological, cognitive and postural control parameters in young athletes.

I think that the manuscript is well developed, the analysis of scientific problem is clear and the paper main propositions is well stated. The statistical analysis and results are comprehensive and satisfying providing some information about the objectives of the study.

The present study has merit and should be shared with the scientific community, however here below my specific comments.

Introduction

Line 64-75: in the previous paragraph the Author have examined the role of psychological and cognitive performance on sport specialization. Please consider to expand the association between postural control and sport specialization

Materials and Methods

Line 144-150: please consider to add, if it is possible, validity and reliability (inter-intra) coefficients regarding the “SAC cognitive screening” and the “BESS” test

Discussion

Line 274-288: please consider to expand the practical applications that practitioners could benefit from the Authors' results

Conclusions

Line 309-310: please consider to report further studies that may arise from the current research

Author Response

1. Line 64-75: in the previous paragraph the Author have examined the role of psychological and cognitive performance on sport specialization. Please consider to expand the association between postural control and sport specialization

Thank you for your valuable comments. We provided some research evidence on postural control among athletes with sport specialization. Please refer to them on line 57 - 61.

2. 

Materials and Methods. Line 144-150: please consider to add, if it is possible, validity and reliability (inter-intra) coefficients regarding the “SAC cognitive screening” and the “BESS” test

Thank you for your constructive comments. We agree with the reviewer that providing the validity and reliability information of the outcome measures will benefit the readers. The validity and reliability regarding the SAC and BESS assessment were added on the outcome measures section. Please refer to them on line 148-149 and 152-153.

3. Discussion. Line 274-288: please consider to expand the practical applications that practitioners could benefit from the Authors' results

Thank you for your constructive feedback. We agree with the reviewer that providing the practical applications will benefit the healthcare professions in addressing sport specialized population. We have added the clinical applications to the healthcare providers based on our study results. Please refer to them on line 335-336.

4. 

Conclusions. Line 309-310: please consider to report further studies that may arise from the current research

Thank you for your constructive feedback. We agree with the reviewer that providing recommendations for future studies will foster the understanding in sport specialization which will benefit the overall health among this population. We have provided several recommendations to future study. Please refer to them on line 356-358.